# Rural Ties and Consumption of Rural Provenance Food Products—Evidence from the Customers of Urban Specialty Stores in Portugal

**DOI:** 10.3390/foods11040547

**Published:** 2022-02-15

**Authors:** Elisabete Figueiredo, Teresa Forte, Celeste Eusébio, Alexandre Silva, Joana Couto

**Affiliations:** 1Department of Social, Political and Territorial Sciences & GOVCOPP—Research Unit on Governance, Competitiveness and Public Policies, University of Aveiro, 3810-193 Aveiro, Portugal; 2Department of Social, Political and Territorial Sciences, University of Aveiro, 3810-193 Aveiro, Portugal; teresaforte@ua.pt (T.F.); joanascouto@ua.pt (J.C.); 3Department of Economics, Management, Industrial Engineering and Tourism & GOVCOPP—Research Unit on Governance, Competitiveness and Public Policies, University of Aveiro, 3810-193 Aveiro, Portugal; celeste.eusebio@ua.pt; 4Instituto de Ciências Sociais, Universidade de Lisboa, Av. Professor Aníbal de Bettencourt, 9, 1600-189 Lisboa, Portugal; alexandre.silva@ics.ulisboa.pt

**Keywords:** consumers motivations, consumers’ images if rural contexts, consumers’ images of rural products, rural provenance foodstuffs, rural ties, urban specialty stores

## Abstract

Consumers’ food preferences increasingly meet concerns of authenticity, health, origin, and sustainability, altogether attributes embodied in rural provenance food products. The dynamics of production, commercialization, and availability of these products in urban centers are growing stronger. This study aims to explore rural provenance food consumption and underlying motivations, the consumers’ images of products and provenance areas, and the influence of rural ties in consumption. Data from a survey directed to 1554 consumers of 24 urban specialty stores located in three Portuguese cities were analyzed. The analysis is based on the differences between frequent and sporadic consumers of Portuguese rural provenance food products. The two groups significantly differ in the reasons provided to acquire the products. Those who buy and consume these products more frequently especially value sensorial features, convenience, national provenance, and the impacts on rural development. Additionally, the motivations to choose rural provenance foods tend to pair with positive images of those products and of their territories of origin. This is intrinsically connected with familiarity, a nuclear notion that encompasses the symbolic images of the products and their origins as actual connections (familiar and otherwise) to rural contexts.

## 1. Introduction

The interest of both consumers and retailers in rural provenance food products has increased in recent years [1,2]. According to Figueiredo [3], rural provenance foods may be defined as all the products whose distinct qualities are anchored in their rural place of production and are shaped by the respective biophysical conditions and food-related cultural traditions. Therefore, the term applies to both officially certified and non-certified products. Often described as local, regional, traditional, authentic, gourmet, or organic [4,5], these products share this common denominator of being interwoven with specific places of origin, production processes and/or cultural features, and traditions [3,4]. 

The growing number (mainly in the last decade) of specialty stores selling rural provenance food in urban centers speaks favorably about new rural–urban connections and fluxes of people, products, capital, and knowledge [1,6]. It is argued that this recent phenomenon may influence the expansion and consolidation of agricultural production in rural areas, thus contributing to rural development, larger economic diversification, and the overall attractiveness of rural territories [3,7,8].

The interest of retailers and consumers [9,10] in provenance foods has been boosted by several policies and strategies [3,11,12] and, as Bowen and Master [13] state, counter the standardizing and industrializing paths of food globalization. Currently, consumers tend to choose rural provenance food products motivated by their sensorial features or guided by personal values or by a perception of those products as more authentic, trustworthy, and, to a certain extent, familiar, which is especially true for national consumers. Familiarity, understood as the knowledge of a product [14,15], may also be related to consumers’ ethnocentrism [16]. Furthermore, in Portugal, as in other southern European countries, as a consequence of relatively recent *de-ruralization* processes, local networks of social relations based on kinship and neighborhood are still quite visible and robust [17]. These complex networks are strongly connected with small-scale agriculture and with the sociocultural characteristics of given social categories that evince strong ties with rural ways of life and sociability, even if the majority of the population is currently living in urban environments. Although further research is needed, it appears that those connections tend to materialize into food habits and access to agri-food products produced by relatives [17]. These may also shape familiarity, knowledge, and preference for rural provenance food products. 

The purpose of this study is to identify and explore the main reasons underlying the valorization and preference for rural provenance foods, the images consumers associate with products and provenance areas, and the influence of rural ties, also identifying the products acquired more in urban specialty food shops. Data from a survey directed to 1554 consumers, which are also clients of 24 Portuguese urban specialty stores located in three Portuguese cities, were analyzed considering the differences between frequent and sporadic consumers of Portuguese rural provenance food products. 

## 2. Literature Review

### 2.1. Urban Specialty Stores Selling Rural Provenance Foods

In the last decade, the number of specialty food stores selling rural provenance products has increased in urban centers. The different identities and roles played by these stores in promoting rural provenance foods and rural development has only recently attracted attention [1,3]. This is guided by the premise that as venues selling foodstuffs with characteristics indissociable of their provenance, these stores may act as showcases of the territories of origin, ways of production, and symbolic dimensions of local, regional, and cultural identities [3]. The rural sociocultural universes of provenance portrayed by these stores are more and more valued by urban consumers whose food preferences are increasingly leaning toward rural provenance food products [4,11,12,18,19]. A case in point is provided by a first attempt to typify urban specialty food stores [1] within the Portuguese cultural context according to the type of rural provenance products sold. The different rural provenance patterns and specialties, as well as overlapping features [1], within and between the found clusters of shops, reveal the diversity of national provenance (especially when addressing overseas customers) as well as their linkages to particular products, specific regions, villages, and producers. 

The manifold definitions of these products may shape different types of stores [1]. Overall, they contribute to fostering consumers’ interest in rural provenance food products by promoting closer communication with customers and creating a specific environment [20]. In this vein, stores selling rural provenance food products coined as local and regional, promote a common appeal to the origin, authenticity, cultural and regional identity, and heritage [3]. In turn, stores selling rural provenance products as gourmet may elevate the authenticity and provenance-distinctiveness of the products by emphasizing their status, exclusivity, and uniqueness, portraying a kind of ‘elite authenticity’ [21]. The differentiation of these stores’ identities and interconnection with the type and nature of products is also shaped by what drives different segments of consumers of rural provenance foodstuffs [22], as discussed in the following sections.

### 2.2. Rural Provenance Food Products’ Choice and Consumption

The theory of consumption values [23] is useful to understand rural provenance food determinants of choice and consumption. These are influenced by functional, social, conditional, epistemic, and emotional values and attributes, revealing extrinsic and intrinsic aspects related both to the consumer and the products [23]. Functional attributes are related to sensorial features (such as taste, flavor, appearance, nutritional content) referred to as crucial motivations to choose traditional and organic foodstuffs [24]. The tangible and material characteristics include appreciation for their flavor [25,26], taste [27,28], and general appearance, including color and sensorial attractiveness. The acceptability of sensorial qualities varies across regional and national contexts being modelled by habits, familiarity, knowledge, beliefs, and culture [29,30].

Conditional aspects embody, among others, elements dependent on the product, personal situation, place, and context [31,32]. The research on convenience factors related to the products and personal situation suggests that more than fostering the consumption of rural provenance products, they may constitute barriers with prices being too high for the available financial resources of the average consumer [33]. This aspect is particularly evident regarding the consumption of organic products [31]. Conditional determinants of place and context are, in turn, related to the selling venues and promotional strategies employed that render these products available, reachable, and appealing. All are perceived as influencing the choice of consumers, albeit with different valences. In this regard, availability was found systematically to be a limitative factor [26,34] speaking unfavorably about the number of venues selling these products in different territories, the production volume, and constancy and, also, of these foodstuffs’ supply chains. The issue of reachability and availability may be mitigated by urban specialty stores’ efforts, also through the products’ promotion to different consumers’ segments, as observed in studies conducted with general and niche retail [35,36]. These stores tend to foster closer relationships and communication with the customer and create particular environments, for which, as concluded by Usitalo [37], stores’ size, personalized customer service, familiarity, and intimacy are perceived as important attributes of convenience. 

Health as a personal attribute may also operate as a conditional value, constituting one of the major determining factors to choose food products [31] alongside well-being, quality of life, and appearance concerns [38]. Birch et al. [39] operationalized consumers’ concerns for personal health, safety, and trust as a typical egoistic or self-interest-based motivation. It was argued that this is driven by the perception of superior intrinsic elements of rural foodstuffs, such as quality, freshness, nutritional value, appearance, trustworthiness, natural, and free of chemical and artificial additives [40]. However, as Birch et al. [39] refer, health concerns may triangulate with social and altruistic motivations, especially if related to affective commitments to further develop the economy and production of one’s national or local context [41]. This includes supporting local communities and producers, fostering more job opportunities, and contributing to rural development and the preservation of the socio-cultural ethos [39] as well as to the protection of the environment and food sustainability in the long run [42].

The promotion of these products, attuned to the need of todays’ consumers to know more about the food they buy and eat [43], uses multiple elements to foster the abovementioned aspects, as well as epistemic and emotional ones. For example, the use of social media, keen on viral or word of mouth and expanding networks, sharing and live updates as well as innovative, attractive, and user-friendly websites [34], may shorten the distance between retailers and consumers. It may also elicit their interest and knowledge about specific products. In fact, a tailored outreach may meet important epistemic factors of choice such as the desire for novelty and knowledge (as well as some emotional drivers such as the satisfaction, joy, pleasure, or happiness of the consumers) [33]. As suggested by Lusk & Briggemen [44], although valuing the products’ attributes themselves, consumers will choose food products if the real or perceived outcomes are in line with their personal values and emotional arousal. 

Among the several determinants underlying the values-based and emotional appeal to consumers, provenance [12] is at the core of a wide set of factors. The more relevant is the perception of authenticity [21,45], naturalness, safety, and trust [40], the superior quality and status of the products [46], and nostalgia [10]. On a slightly different note lies an ethics of care, embodied by concerns over animal welfare, food sustainability, and ecological footprint [47]. This linkage to provenance is often enacted by consumers through a national/regional ethnocentrism of choice [16]. The concept initially employed to depict normative beliefs towards preferring national products (over imported ones), now also encompasses preferences at a sub-national level. The ethnocentric trends target local, regional, and traditional food products, the latter being perceived as distinct, since the identification of a particular origin, although not sufficient, is a sine qua non condition to qualify a product as traditional. The way these factors are understood, and thus the guiding choices, may vary across cultures (as exemplified by Amilien et al. [48] comparing French consumers’ perceptions and preference for local food quality with Norwegian ones) but are often based on sensorial features alongside nostalgia. The differences at this level have been recently tackled by a growing stream of research on consumer nationalism [46,49], a concept that is evolving from describing efforts and enacting a political statement boycotting anything foreign to being used as describing consumers’ positive attachment to national, regional, and local food products.

### 2.3. Rural Ties as Explaining Consumers’ Food Choices

High-quality products, when produced in rural contexts, may elicit a sentimental longing for the past [50] related to an emulation of authentic nostalgic values evoked by rural-based individual and family history and identity, as well as local culture [26]. As a determining factor of choice, the emotional dimension of national in-group belonging may be strengthened by the coherence with a values-based choice, countering globalized markets and unsustainable agri-food systems [11,13] through supporting local and more sustainable productions and supply chains [19]. This seems to have gained further relevance after supply chains’ failures during the COVID-19 pandemic, which could raise support for food sovereignty and shorter supply chains.

The perception of rural provenance products as ambassadors of local, national, and cultural heritage and identity [51] is shown to even surpass other important convenience determinants of choice, such as its cost or hedonic nature, as is the sensorial appeal [32,39]. This is especially true for national consumers with rural (mainly family-based) ties who are consequently more knowledgeable about rural provenance foods [15]. 

Food, as expressed by Bardone and Spalvena [49] (p. 43), is an important vehicle of cultural identity, playing a relevant role in “authenticating an ethnic or national culture”. Accordingly, rural provenance foods are often portrayed by consumers (as well as by promotion strategies) as pre-industrial and pre-capitalist, related to peasant societies and to particular territories and their festivities, habits, and culture [3]. These representations follow the transformations of rural territories, especially in southern European countries where rural areas are increasingly seen as multifunctional spaces [3,52]. As evidenced by Bessière [52], Fonte [4], and Fonte and Papadopoulos [11], rural identity has been redefined through those dynamics of change, especially those related to rural culture and rural elements’ commodification processes. These match a growing interest in rurality and foster new practices and social demands in which rural provenance food represent an important part [19], given its role in reconnecting consumers to specific (rural) places of production, inviting them to be part of an (often) already lost (or changed) rural culture and identity [52]. 

To consume rural provenance food may thus be fostering the preservation of traditions, habits, and cultural heritages [3], while at the same time contributing to rural development and rural and agricultural sustainability [13]. These are also important values underlying current consumers’ preferences regarding food products. If, on the one hand, the desire to preserve traditional rural foods and rural territories of origin has been used to justify political and policy choices and food labelling related decisions [12], on the other hand (and perhaps related), it is linked to nostalgia, familiarity, and other positive feelings towards rurality [3,19]. The contribution of these processes to the sustainable development of the rural territories of provenance [3,4,11,18,19] seems particularly relevant in southern European countries such as Portugal, characterized by persistent dynamics of rural marginalization [3,11] following relatively recent *de-ruralization* paths. However, it is precisely this recency that may explain the persistence of strong ties and complex social links with rural territories, small-scale, and traditional agricultural productions mainly based on family relationships [17] and, to a certain extent, the knowledge, experience, and familiarity [15] underlying the choices for rural provenance food products. 

Despite the timeliness of the topic, the literature connecting traditional food products consumption choices and practices with rural family ties, as well as the role of food in maintaining those ties, is not abundant [17].

## 3. Materials and Methods

### 3.1. Data Collection

To identify and analyze the main reasons underlying the preference for Portuguese rural provenance food products and unveil the differences between frequent and sporadic consumers, a survey was conducted between October 2020 and June 2021 on 1554 clients of 24 urban specialty food stores located in three Portuguese cities, Aveiro (*n* = 5), Porto (*n* = 10), and Lisbon (*n* = 9). 

The 24 stores were randomly selected (using a table of random numbers) based on a hierarchical cluster analysis resulting from a previous survey targeting stores’ owners (*n* = 113). Data of this survey were used to segment the stores according to the criteria of selling rural provenance food from Portugal and being of small to medium size dimensions. Three clusters were identified based on the main products sold by the stores: (i) ‘The Wine Focused’, with stores selling wine and other beverages; (ii) ‘The Rural Provenance Focused’ with stores selling regional and rural food products and (iii) ‘The Generalist’, encompassing several products from a wide variety of regions (see Silva et al. (2021) for a thorough characterization of these clusters). 

The questionnaire (see Appendix) was elaborated both in Portuguese and English and, based on the literature review, addressed the consumption of traditional rural national-based products [5,16,23], the products acquired at the store on the survey date [3], the frequency of consumption, region of origin [9,12], the reasons to select Portuguese rural provenance products [18,39,46], as well as the images of rural territories and traditional food products [51,53]. A pilot test was carried out to customers from stores in Porto and Aveiro (*n* = 10), whose inputs (mainly regarding language simplification and introduction of some alternative responses in the open-ended questions) were included in the final script of the questionnaire.

### 3.2. Data Analysis

Data were analyzed using the software SPSS (*Statistical Package for Social Sciences)*, version 25 (IBM, USA). The sample was divided according to the responses to the dichotomous question ‘*Do you usually consume traditional food products of Portuguese rural origin?*’, resulting in a group with frequent consumers (*n* = 1175) and another group of sporadic consumers (*n* = 369). Both frequent and sporadic consumers are clients of at least one of the 24 stores considered. To compare these two groups, Chi-square tests were used for qualitative variables, namely sociodemographic characteristics (gender, age, marital status, education level, economic status, nationality, monthly household income); familiarity with rural areas, assessed by seven dichotomous items (e.g., relatives living in Portuguese rural areas; visited rural areas in the last three years; visited rural areas to buy and/or consume food products); the type of products bought in the date of the survey and their regions of origin; image of rural territories and image of rural provenance food products. In addition, to compare the two groups on the importance attributed to a set of motivations to prefer and acquire Portuguese rural food products (assessed through the Likert Scale—from 1 = not important to 5 = very important), independent samples T-tests were used.

## 4. Results and Discussion 

### 4.1. Sample Profile

The sample was analyzed considering the customers of the urban specialty stores that frequently consume Portuguese rural provenance food products (76.1%) vis-à-vis those who do not consume these products often (23.9%). As shown in Table 1, the proportion of female participants buying rural provenance products at the stores, at the date of the survey, is slightly higher than males, but no significant differences in this respect between the two groups were found. Likewise, no significant differences were found regarding marital status, education levels, monthly household income, and economic status. The majority of respondents in both groups are married (60.4%), have completed higher education (50.6%), are employed (62.7%), and have a monthly household income below 2200 € (76.2%). The lack of significant differences considering these variables suggests that, in the sample, the socioeconomic condition per se is not influencing the frequent or sporadic consumption of the respondents. 

As shown in Table 1, the majority of respondents (73.4%) are between 25 and 64 years old. Differences between cohorts show that older consumers acquire this type of product more frequently than younger ones, particularly those younger than 25 years old. The particular preference of older consumers for products anchored in Portuguese rural areas may also reflect a higher knowledge, experience, and familiarity (in line with Seo et al. [15]) and, therefore, an evocation of a sentimental longing and nostalgia (as also put forward by Truninger [10] and Sedikides et al. [50]) further explored below regarding consumers’ images of rural territories and food products, as well as the reasons to acquire these products.

Even though the majority of respondents are Portuguese (75.8% of the sample), a significant difference between these respondents and non-Portuguese customers is evident. Portuguese respondents are more likely to belong to the group of those who acquire and consume rural provenance food products more often, while the majority of non-Portuguese customers do not. This difference may be explained by convenience-related factors, namely proximity and availability (as stressed by Carolan [35] and Toften and Hammervoll [36]), as well as by awareness and acquaintance with the products and the selling venues (as explained also by Seo et al. [15] and Camillo and Di Pietro [32]). On the other hand, the apparent higher valorization of national (instead of imported), regional, and local food products by Portuguese consumers may result from a possible growing trend of a national/regional ethnocentrism [16], usually reflecting consumers’ normative beliefs about the better nature of these products allied to a positive attachment, in line with the debate of Bardone and Spalvena [49]. As will be further explored, this attachment may spill over and not circumscribe to products themselves but to their wider material and symbolic relations with the places of origin. 

### 4.2. Rural Provenance Food Products Acquired, Motivations and Images

#### 4.2.1. Type of Products

Crossing with the analysis of products bought at the shop when surveyed (Table 2), few significant differences were found. In fact, the respondents in the two groups mostly acquire wine and other beverages, cheese, and other milk derivatives, and cured meat and other animal-based products. These correspond to the food products generally identified with Portugal and, therefore, as also stressed by Figueiredo [3], the main products both sold by and bought in urban specialty food stores. Despite the homogeneity between the two groups regarding the products acquired, a significant difference is found between those who buy vegetables, fruits, and derivatives that tend to be more frequent consumers of these products, and those who bought sweets and cosmetics and similar products. This difference may be explained by the perishable character of the first type of products which encourages more frequent purchases and consumption, as well as by the fact that these food products of Portuguese rural origin are amongst the more typical ones.

As for the geographical origin of the products, the only significant difference between the two groups relates to the fact that products from *Beira Litoral* (center region of the country) are more likely to be bought by consumers who do not regularly buy Portuguese rural provenance foodstuffs. This may be related to the type of products bought and to the fact that this region is not, especially when compared with the main provenances of *Trás-os-Montes* (North region), *Beira Interior* (Centre region), and *Alentejo* (South region), as important in the production of wine, cheese, or cured meat products.

#### 4.2.2. Acquisition and Consumption Motivations

Independent samples T-tests were conducted to identify the reasons more associated with acquiring more often and consuming rural provenance products (Table 3). Significant differences were found concerning the motivations related to national provenance—‘*That they are produced in Portugal*’ and ‘*To be produced in Portuguese rural areas’—*all presented by customers who would more likely belong to the group of frequent buyers and consumers of provenance food products. These motivations may also be aligned with consumers’ nationalism (as pointed out by Bardone and Spalvena [49]) or ethnocentrism (as explained by Fernández-Ferrín et al. [16]) and explained by the larger number of Portuguese consumers amongst the respondents. Even though this preference for national rural food products may enact a nostalgic longing for the past, often related to the referred patriotic feelings, it may also be the expression of an affective commitment (as evinced by Memery et al. [41]) to further develop the economy and production of one’s national or local context and help local communities and producers (also expressed in the item ‘*To support Portuguese agriculture and rural areas*’). This could be objectively achieved through the creation of more job opportunities, revitalization of mainland areas and contribution to rural development, and the preservation of a sociocultural ethos [48]. Whereas the first draws on an emotional-based appeal, the second, more future-oriented, combines protection for one’s own ingroup with an ethics of care towards the environment (in line with DuPuis and Goodman [42] and Amilien et al. [48]), rural areas, and their social capital much affected by the dynamics of rural marginalization [3].

Although less important, ‘*To know the producers*’ is also a motivation to buy rural provenance foods amongst those frequent consumers. This may be related to a higher familiarity with the products and to a greater knowledge of both products and producers, as evinced by Castelló and Mihelj [46], Johnson and Russo [14], and Seo et al. [15].

The same pattern occurs in those frequent buyers and consumers who value the sensorial features of the products, such as taste and their healthier nature, as fundamental determinants of choice. These findings are in line with the studies of Castellini et al. [27] and Von Meyer et al. [28] on the relevance of taste. These findings are also coherent with Kushwah et al. [31] about the significance of health concerns and values as key determinants of food choice and, as our evidence suggests, factors of differentiation between frequent and non-frequent consumers. 

As shown in Table 3, additional significant differences between the two groups relate to the characteristics of the sale and not as much to the product or the producer. In fact, aspects such as ‘*having a fair price’*, ‘*to trust in the store and in its specialized customer service*’, together with ‘*the fact that I can buy the products in my residency area*’ are more valued by frequent consumers. The relevance attributed to a fair price is in line with the usually higher price of this type of product, as pointed out by Jansen [33] and, for the particular case of organic foods, by Kushwah et al. [31].

Product availability near the consumers’ residency areas is also pointed out (for example by Bryla [26] and Barska and Solis [34]) to be an important determinant of food choice. By making available rural provenance products from different regions of origin in city centers, urban specialty stores enable their acquisition and consumption, promoting closer connections between the rural places of production and the (urban) places of consumption [1,3]. These stores’ efforts regarding reachability and availability, are also visible through the products’ promotion to different consumers’ categories, as observed by Carolan [31] and Toften and Hammervoll [36], namely, as in our sample, to frequent and non-frequent buyers. The latter, as shown in Table 3, tend to value more the fact that the products are advertised on mass media and/or through social media networks very often used by the specialty stores as a means of promotion. As stressed by Barska and Solis [34] among others, social media may contribute to shortening the distance between retailers and consumers and promote interest and knowledge about specific products, especially for consumers less used to buying and consuming rural provenance foods. Therefore, this result suggests that social media, with its potential in creating networks, may be actually reaching more unaware consumers.

#### 4.2.3. Images about Products and Rural Areas of Provenance

Consumers’ images of Portuguese rural areas were also analyzed to identify their relevance and influence in the likability to consume food products of Portuguese rural origin. Respondents in both groups characterize Portuguese rural areas through mainly positive elements, such as ‘*gaze, tranquility and well-being*’, in line with the discussion undertaken by Soares da Silva et al. [53] as well as with its identification with ‘*environment and natural elements*’ (as suggested, among others, by Figueiredo [54]). However, as shown in Table 4, significant differences were found between the two groups analyzed, regarding the images of Portuguese rural areas, on the one hand, as abandoned, isolated, and ageing and, on the other hand, as the places in which food products and their characteristics are anchored. Consumers who characterize rural territories as abandoned, isolated, and ageing are more likely to belong to the group that does not consume rural provenance food products frequently. This suggests that a negative image of the areas of origin may somehow make the products less attractive and impede their consumption. Conversely, those who identify rural areas with the products themselves and their distinct qualities are more likely to belong to the group that buys rural provenance foodstuffs frequently. 

Both results, in different ways, suggest that the valence and a given content of an image of rural contexts are related to the interest and choice of rural foodstuffs, reinforcing the degree to which these products are rooted in their provenance. The first images show that a grim representation of rural provenance translates to a lack of interest in what comes from it, either by ignorance of the existence of products or by a certain spillover of the negative valence of the context itself. The second image suggests that rural provenance is also perceived by many as interwoven with an idea of food production and of the distinct qualities of place and food (in line with Guerrero et al. [51]) and in a not so simple way since it refers to a wide range of food products and respective features. This may also be related to the frequent consumers’ knowledge, experience, and familiarity of both rural territories and food products (as referred also by Seo et al. [15]). The characterization of rural areas as the places of provenance and distinction of food products also suggests a kind of collective property grounded on given know-how and tradition. These are responsible for the inherent quality of this type of product, forged across generations, each reinterpreting its traditional value and typicity (as emphasized by Figueiredo [3]).

Considering all the results, even without statistical differences, the characterization of rural areas as authentic, traditional, and unique is slightly higher by those who consume these products frequently. This may suggest an overlap between these descriptors and what is searched in the deriving foodstuffs. Authenticity features, one of the most important determinants of rural provenance food consumption (as stressed by Lacoeuilhe and Lombart [45]), are often related to a concern with the unique qualities of the food products shaped by the biophysical and cultural features of the places of production. The search for products anchoring on a traditional provenance may again be understood in the light of national/regional ethnocentrism in food choice (as highlighted by Fernández-Ferrín et al. [16]) or consumer nationalism (as in Castelló and Mihelj [46]) once more strongly connected with familiarity and knowledge about foodstuffs and their places of origin. 

Interestingly, despite the differences already discussed, the images of rural provenance products held by both groups are fairly homogeneous (Table 5). Apart from the fact that not so frequent consumers are more likely to emphasize their ‘*trustworthy*’ character, which is also a sign of a favorable widespread image, there is indeed a great consensus regarding the characterization and description of those products, evincing is the absence of a clear association between specific images and the frequency of acquisition and consumption.

Both groups analyzed show specific images of Portuguese rural provenance food products coherent with the known determinants of food choice. Their general quality is a common image, revealing the relevance of the concomitant products’ features for the consumers, as also concluded by Birch et al. [39] and Andersson [40]. This general image is followed by the characterization of rural provenance food products based on their sensorial features (evidenced in the use of words referring to taste, flavor, and appearance, among others). These are amongst the main motivations to choose traditional foods and together with health concerns, organic foodstuffs, as Sidali et al. [24] refer. 

Respondents in both groups also associate rural provenance foodstuffs with specific products, mainly within the categories of meat-related products, cheese, and vegetables (Table 5), the first two corresponding to the most typical rural provenance foodstuffs in Portugal. This may be related to the characterization of these products as traditional, hand-made, with a production based on experience and know-how, which is another frequent image held by both groups, and alongside ‘*distinction and authenticity’*, corresponding to important determinants of rural provenance foodstuffs ([21,45]). In the same vein, the characterization of these products as related to nature, environment, and sustainability aspects meet both the altruistic motivations (as stressed by Birch et al. [39]) underlying their acquisition, as health and safety concerns [40]. Interestingly, price is the least evoked element in both groups’ images of the products.

#### 4.2.4. Rural Ties as Determinants of Food Choice

Significant differences between the two groups were found in all the variables related to familiarity and connections with rural areas, visiting habits, products, and activities (Table 6). Overall, the results indicate that consumers who have an interest in and connection with rural areas are more likely to belong to the group of frequent buyers and consumers of rural provenance food products than those who do not. As shown in Table 6, these rural ties vary in degree of socialization and connection, ranging from having relatives living in rural contexts and visiting them to having visited these areas in the last three years. These visits served specifically to buy and taste local gastronomy and food products, as well as to participate in local traditional economic activities.

Portugal may be characterized by a recent process of *de-ruralization* contributing to still strong connections and linkages to rural territories, either through family relations [17] or other personal networks. National consumers who nurture rural linkages (often family-based, as shown in Table 6) are more prone to be knowledgeable about rural provenance foods [15]. This knowledge is a key element of familiarity with the products and producers, revealing that rural ties may model and impact the appeal, purchase, and consumption of these products. Being socialized within a certain context, ranging from familiar habits and beliefs to local, regional, or national culture is known to impact food preferences. For instance, the cultural context of upbringing is known to influence consumers’ acceptability of sensory qualities of rural provenance foods [29] which often translates into a higher valorization and appreciation of the more authentic and sensorial complex features that characterize these products. It also appears to strengthen the persuasiveness of the emotional appeal (of the products and to the consumers) of stability, permanence, and trustworthiness communicated by known sources, places, and traditions [43,50].

To be physically and symbolically close to rural contexts and communities also brings some awareness and empathy towards these realities’ dynamics and necessities [39,42]. One may argue that this engagement is more personal when the family roots are stronger, becoming political, sometimes, especially when sustainable socio-economic development of these areas is at stake. The emotional and values-based appeal of choosing these products instead of those that result from massified agri-food systems [4,11,13] may be, in fact, stronger when the support for local and more sustainable supply chains [19,42] is driven by an actual and symbolic connection and ties with rural communities. These may be powerful in evoking emotions of nostalgia and sentimental longing for the past (as stressed by Sedikides et al. [50]) paired with an ethics of care. The usually soothing element of familiarity, at a sub-conscious level, carries many socio-cultural meanings and roots that activate sensorial acceptability and preferences, pleasant memories, and feelings (as stressed by Bryla [26]), and in-group belonging and protection. 

The evidence presented in Table 6 also suggests that rural provenance foods are important reflections of a given culture and territory (in line with Bardone and Spalvena [49]), therefore playing a significant part in local, regional, and national identities and connections. Rural provenance food products are also seen as related to cultural manifestations, as specific festivities and habits [3] that contribute to forming and consolidating cultural identities. These rural-based sociocultural elements, being important for every region and country, seem to assume particular relevance in Portugal and other southern European countries wherein long-lasting processes and dynamics of rural marginalization exist [3,11]. This endogenous focus reaches the highest ideological ground in national/regional/local ethnocentrism (in line with Fernández-Ferrín et al. [16]) or consumer nationalism (as in Castelló and Mihelj [46]) to which familiarity is key. Overall, these results indicate that the preference for and acquisition and consumption of rural foodstuffs is associated with proximity and connections with their territories of provenance, corroborating the power of familiarity in forging taste, preferences, and dietary habits [30].

## 5. Conclusions

Despite the extensive literature on food consumption determinants, the preference for rural provenance, traditional, nationally-produced food products is a relatively overlooked topic, particularly when considering the differences between frequent and sporadic consumers. This study aimed at contributing to understanding the interdependence of those differences with the motivations in acquiring, consuming, and valuing rural provenance food products, consumers’ images on both food products and territories of provenance, and the existing ties with rural territories. 

Taking as a starting point the customers of urban specialty stores selling rural provenance food products and analyzing their frequent or sporadic consumption of those products, our results strongly suggest the nuclear role of those stores in promoting the products and their territories of origin. This may have an important impact on the consumption of rural provenance foods in (re)shaping rural–urban connections and in fostering sustainable agriculture and rural development. 

Specifically, our results highlight the overlap between the images of urban specialty food stores’ customers regarding rural territories and rural food products and their motivations and criteria of food choice. This corroborates the strong interconnection of these products with their regions of provenance, up to an extent of symbolic spillover between the characteristics of the products, the processes of production, and the territories of origin, with features of one element being extensible to perceive the other. In fact, despite consumers’ images of Portuguese rural territories being generally positive, there is a strong association between negative images of rural areas (as abandoned, isolated, ageing) and a lower frequency in the acquisition of rural provenance products. Conversely, more positive images of rural areas, particularly as places of food provenance, are associated with a higher acquisition of food products. 

Rural provenance food products are generally characterized in a very positive manner by both groups of consumers analyzed here. As shown, both groups possess images of rural provenance foods that are aligned with the discussed determinants of food choice, namely their trustworthy character, the emphasis on sensorial features, their authenticity and distinction, and their general quality. These images seem to be strongly interconnected to the motivations for the acquisition and consumption of rural provenance foods. Significant associations were found regarding motivations related to the national provenance of the food products and the higher frequency of acquisition. This is suggestive of ethnocentrism and nationalism as powerful drivers and determinants of food choice and consumption. The preference for national rural food products may also be the expression of an affective commitment to contribute to the preservation and sustainable development of those productions and their territories of origin. It is also found in the more common images of rural areas and food products amongst frequent consumers, probably related to familiarity with rural territories and to concerns of sustainable food production and consumption. The latter seems to motivate the support to local communities and producers, more visible amongst frequent consumers which are also concerned with environmental protection and national agriculture and rural territories’ development. Even though these are increasingly widespread food consumption-related concerns and motivations, they seem particularly important in southern European countries such as Portugal, in which rural marginalization dynamics have been persistent and difficult to overcome. Finally, different ties and degrees of familiarity with Portuguese rural territories—from blood liaisons to knowledgeably consuming the products or often visiting rural areas—emerged from our results as the most important determinants of food choice. Notable differences were found, regarding familiarity with Portuguese rural territories, with frequent consumers presenting closer and stronger connections with those territories. Again, the strong ties with rural territories and agricultural productions revealed by urban populations are still quite evident in Portugal, as in other Southern European countries, as a result of relatively recent *de-ruralization* dynamics, the persistence of local networks of social interactions based on kinship and neighborhood relationships. These complex processes evinced stronger connections with rural ways of life, habits, practices, and values in these countries, vis-à-vis nations that experienced earlier urbanization processes. The stronger ties with rural territories also relate to consumers’ ethnocentrism and nationalism regarding food consumption and acquisition motivations and practices. They are also related to familiarity, knowledge, and experience regarding rural provenance food products that motivate their consumption and acquisition. 

Showing that knowledge of products and regions are linked to buying frequency, our results suggest that sellers can benefit from more knowledgeable buyers, thus presenting a rationale for seller involvement in delivering this information to potential customers. Our results also suggest that there could be a mutual reinforcement, both with positive and negative directions, between how the rural is perceived and the demand for rural products which could be further explored in future research. From a policy perspective, the support and development of good practices in sustainable food production could be increased through this link between demand and images of rural regions by helping producers and regional stakeholders to develop closer associations between the promotion of territories and sustainable production processes.

Notwithstanding its theoretical and practical contributions and the salience of the topic, one limitation of this study is that it was conducted in just one country, making it difficult to extrapolate results to different regions and countries. A more thorough discussion could be developed by comparing contexts with contrasting degrees of rural marginalization, histories of *de-ruralization*, and different degrees of urban-rural kinship linkages. Furthermore, within Portugal, just a sample of urban specialty food stores from only three cities was analyzed. Even though two of those cities (Porto and Lisbon) are the country’s most populated and touristic, it is difficult to extend the results to other cities and towns within the country with diverse characteristics. Therefore, similar research in other countries and cities with different characteristics would be useful for comparative studies and further analysis of the influence of diverse contexts regarding the preference and choice of (local) rural provenance foods. Considering less urbanized contexts would perhaps unveil other types of ties and connections between consumers, food, and territories of provenance. 

Further research would also benefit to explore familiarity variables in articulation with other aspects of interest, namely political ideology and values in line with the new stream of consumer nationalism and ethnocentrism that is gradually perceiving it in a less segregationist and more empowering light. 

## Figures and Tables

**Table 1 foods-11-00547-t001:** Sample Profile.

Profile	Total		Consumption of Traditional Food Products of Portuguese Rural Origin *		Chi-Square Test	
	*n*	%	Consume Frequently	Does not Consume Frequently	Value	*p*-Value
			(*n* = 1175, 76.1%)	(*n* = 369, 23.9%)		
**Gender**						
Male	756	49.1	50.2%	45.5%	2.420	0.120
Female	785	50.9	49.8%	54.5%		
**Age**						
Less than 25	101	6.5	5.6%	**9.5%**		
[25–64]	1133	73.4	73.4%	73.4%	**8.603**	**0.014**
More than 64	310	20.1	**21.0%**	17.1%		
**Marital status**						
Single	428	27.9	27.7%	28.5%		
Married/Cohabiting	927	60.4	60.1%	61.1%	1.567	0.667
Divorced	109	7.1	7.2%	6.8%		
Widowed	72	4.7	5.1%	3.5%		
**Education level**						
Less than secondary education	425	27.7	27.8%	27.4%		
Secondary education	332	21.7	20.9%	24.2%	1.899	0.387
Higher education	775	50.6	51.3%	48.4%		
**Economic status**						
Employed	965	62.7	62.5%	63.6%		
Student	103	6.7	6.1%	8.7%		
Retired	339	22.0	22.6%	20.1%	4.311	0.366
Unemployed	101	6.6	6.8%	5.7%		
Other	30	2.0	2.0%	1.9%		
**Nationality**						
Portuguese	1166	75.8	**78.5%**	67.1%	**19.680**	**0.000**
Non-Portuguese	373	24.2	21.4%	**32.9%**		
**Monthly household income**						
Less than 1000 €	445	40.6	39.1%	45.4%		
[1001–2200 €]	390	35.6	37.5%	29.8%	5.863	0.118
[2201–3000 €]	97	8.9	9.0%	8.4%		
More than 3000 €	163	14.9	14.4%	16.4%		

* Percentage in columns. Values in bold correspond to the highest values when statistically significant differences exist.

**Table 2 foods-11-00547-t002:** Type of products acquired and regions of provenance.

Products bought	Total		Consumption of Traditional Food Products of Portuguese Rural Origin *		Chi-Square Test	
	*n*	%	Consume Frequently	Does not Consume Frequently	Value	*p*-Value
			(*n* = 1175, 76.1%)	(*n* = 369, 23.9%)		
**Type of products ****						
Wine and other beverages	463	30.3	30.2%	30.7%	0.034	0.853
Cheese and other milk derivatives	335	21.9	23.0%	18.8%	2.878	0.090
Cured meat and other animal-based products	326	21.3	21.3%	21.5%	0.004	0.945
Vegetables, fruits, and derivates	300	19.6	**20.9%**	15.5%	**5.277**	**0.022**
Sweets	170	11.1	10.0%	**14.7%**	**6.145**	**0.013**
Bread and cereal products	157	10.3	10.4%	9.8%	0.131	0.718
Honey, jams, and preserves	104	6.8	6.6%	7.6%	0.486	0.486
Olive oil	100	6.5	7.1%	4.9%	2.176	0.140
Crafts and similar products	32	2.1	2.0%	2.4%	0.290	0.591
Hygiene, cosmetics, and similar products	26	1.7	1.2%	**3.3%**	**7.033**	**0.008**
**Origin—Agricultural regions ****						
*Trás-os-Montes*	598	41.8	42.5%	39.6%	0.984	0.321
*Beira Interior*	398	27.8	26.6%	31.5%	3.204	0.073
*Alentejo*	235	16.4	17.0%	14.8%	0.961	0.327
*Entre Douro e Minho*	228	15.9	16.0%	15.6%	0.040	0.842
*Beira Litoral*	138	9.6	8.7%	**12.5%**	**4.597**	**0.032**
*Ribatejo e Oeste*	84	5.9	5.9%	5.8%	0.000	0.985
*Algarve*	10.00	0.70	0.01	0.01	(a)	

* Percentage in columns. Values in bold correspond to the highest values when statistically significant differences exist. ** Only the values corresponding to “yes” are presented. (a) The assumption of the Chi-square test was not observed.

**Table 3 foods-11-00547-t003:** Reasons to buy Portuguese rural provenance products.

Reasons to Buy Portuguese Rural Food Products *	Total		Consumption of Traditional Food Products of Portuguese Rural Origin *		*T-*Test	
	*n*	Mean	Consume Frequently	Does not Consume Frequently	Value	*p*-Value
			(*n* = 1175, 76.1%)	(*n* = 369, 23.9%)		
			Mean	Mean		
**That they are produced in** **Portugal**	**1553**	**4.25**	**4.34**	3.98	**6.782**	**0.000**
If they look good	1552	4.09	4.08	4.11	−0.503	0.615
That they are local	1475	4.12	4.14	4.07	1.353	0.176
Having a fair price	1553	4.19	**4.23**	4.07	**2.895**	**0.004**
If they taste better	1552	4.34	**4.38**	4.22	**3.172**	**0.002**
If they are fresh produce	1552	4.18	4.20	4.12	1.431	0.153
If they are officially certified (PDO, IGP, Organic…)	1551	3.50	3.52	3.43	1.304	0.193
To know the producers	1551	3.47	**3.51**	3.32	**2.831**	**0.005**
To know the products’ brands	1550	3.50	3.52	3.44	1.254	0.210
To know the products already	1549	3.64	3.66	3.56	1.566	0.118
If they have been recommended by friends and/or family	1551	3.78	3.78	3.78	0.104	0.917
If they are small-scale produced	1549	3.83	3.87	3.75	1.865	0.062
To be produced in Portuguese rural areas	1550	3.96	**4.02**	3.80	**3.532**	**0.000**
The fact that I can buy the products in my residence area	1544	3.64	**3.73**	3.35	**4.969**	**0.000**
Being advertised on mass media/ social media	1550	3.18	3.13	**3.38**	**−3.229**	**0.001**
That they are healthier	1549	3.92	**3.98**	3.74	**3.856**	**0.000**
To trust in the store and in its specialized costumer service	1549	4.02	**4.10**	3.79	**5.106**	**0.000**
To support Portuguese agriculture and rural areas	1551	4.11	**4.16**	3.94	**3.631**	**0.000**
Their nutritional information	1548	3.79	3.81	3.73	1.202	0.230
If their production carries a low environmental impact	1552	3.95	3.98	3.85	1.870	0.062

* Items classified in a five-point type Likert scale from 1, “ less important”, to 5, “more important”. Values in bold correspond to the highest values when statistically significant differences exist.

**Table 4 foods-11-00547-t004:** Consumers’ images of Portuguese rural areas.

Image of Portuguese Rural Areas	Total		Consumption of Traditional Food Products of Portuguese Rural Origin *		Chi-Square Test	
	*n*	%	Consume Frequently	Does not Consume Frequently	Value	*p-*Value
			(*n* = 1175, 76.1%)	(*n* = 369, 23.9%)		
**Words related to rural** **Areas ****						
Gaze, tranquility, and well-being	571	37.0	36.3%	39.3%	1.068	0.301
Environment and natural elements	461	29.9	29.3%	31.7%	0.759	0.384
Farming	270	17.5	18.2%	15.4%	1.429	0.232
Abandonment, isolation, and ageing	254	16.5	15.2%	**20.6%**	**5.996**	**0.014**
Roots and nostalgia	246	16.0	15.9%	16.3%	0.034	0.854
Authentic, traditional, and unique	233	15.1	15.7%	13.3%	1.268	0.260
Places, villages, and ways of life	206	13.4	13.2%	13.8%	0.089	0.765
Food products and characteristics	184	11.9	**13.0%**	8.7%	**4.907**	**0.027**
Undeveloped and problematic	83	5.4	5.0%	6.5%	1.198	0.274
Growth and diversity	50	3.2	3.6%	2.2%	1.785	0.182

* Percentage in column. Values in bold correspond to the highest values when statistically significant differences exist. ** Only the values corresponding to “yes” are presented. (The categories presented in Table are the result of the grouping of the words used spontaneously (see Question 2 of the Questionnaire in Appendix A) by the respondents to describe Portuguese rural areas into 10 dichotomic variables).

**Table 5 foods-11-00547-t005:** Consumers’ images of Portuguese rural provenance products.

Image of Portuguese Rural food Products	Total		Consumption of Traditional Food Products of Portuguese Rural Origin *		Chi-Square Test	
	*n*	%	Consume Frequently	Does not Consume Frequently	Value	*p-*Value
			(*n* = 1175, 76.1%)	(*n* = 369, 23.9%)		
**Words related to rural food products ****						
General quality	329	21.8	21.2%	23.8%	1.081	0.298
Sensorial features of products	312	20.7	19.9%	23.2%	1.892	0.169
Meat and animal-based products	297	19.7	19.3%	21.0%	0.551	0.458
Cheese and other milk derivates	252	16.7	16.9%	16.1%	0.121	0.728
Hand-made, traditional, experience and know-how	180	11.9	11.2%	14.2%	2.372	0.142
Vegetables cereals and fruits	178	11.8	11.9%	11.5%	0.050	0.823
Distinction and authenticity	177	11.7	11.5%	12.6%	0.322	0.570
Nature/environment/sustainability	174	11.5	10.9%	13.4%	1.620	0.203
Wine	144	9.5	9.6%	9.3%	0.038	0.846
Farmers and farming	112	7.4	8.1%	5.2%	3.514	0.061
Honey, jams, and sweets	76	5.0	4.7%	6.0%	0.952	0.329
Family/nostalgia	74	4.9	5.4%	3.3%	2.747	0.097
Freshness	59	3.9	3.9%	3.8%	0.010	0.921
Organic	55	3.6	4.0%	2.5%	1.942	0.163
Selection/monotony	52	3.4	3.9%	1.9%	3.424	0.064
Regional, local, from specific places	47	3.1	3.6%	1.6%	3.493	0.062
Trustworthy	33	2.2	1.8%	**3.6%**	**4.198**	**0.040**
Gastronomy and cuisine	29	1.9	2.2%	1.1%	1.766	0.184
National character	22	1.5	1.6%	1.1%	0.450	0.502
Chemically free/healthy	21	4.7	4.9%	4.1%	0.401	0.527
Price	12	0.8	1.0%	0.3%	(a)	

* Percentage in column. Values in bold correspond to the highest values when statistically significant differences exist. ** Only the values corresponding to “yes” are presented. (a) The assumption of the Chi-square test was not observed. (The categories presented in Table are the result of the grouping of the words used spontaneously (see Question 3 of the Questionnaire in the Appendix A) by the respondents to describe Portuguese rural provenance food products into 20 dichotomic variables).

**Table 6 foods-11-00547-t006:** Familiarity with rural areas.

Familiarity with Rural Areas	Total		Consumption of Traditional Food Products of Portuguese Rural Origin *		Chi-Square Test	
	*n*	%	Consume Frequently	Does not Consume Frequently	Value	*p*-Value
			(*n* = 1175, 76.1%)	(*n* = 369, 23.9%)		
**Relatives living in Portuguese rural areas**						
Yes	775	50.4	**52.1%**	44.7%	**6.180**	**0.013**
No	764	49.6	47.9%	**55.3%**		
**Visited rural areas in the last three years**						
Yes	936	60.7	**63.7%**	51.2%	**18.278**	**0.000**
No	606	39.3	36.3%	**48.8%**		
**Visited relatives in rural areas in the last three years**						
Yes	396	25.6	**28.0%**	18.2%	**14.266**	**0.000**
No	1148	74.4	72.0%	**81.8%**		
**Visited rural areas to taste local gastronomy and wines**						
Yes	767	49.7	**51.8%**	42.8%	**9.122**	**0.003**
No	777	50.3	48.2%	**57.2%**		
**Visited rural areas to buy food products**						
Yes	566	36.7	**38.9%**	29.5%	**10.582**	**0.001**
No	978	63.3	61.1%	**70.5%**		
**Visited rural areas to buy handicraft**						
Yes	501	32.4	32.3%	33.1%	0.083	0.773
No	1043	67.6	67.7%	66.9%		
**Visited rural areas to participate in local traditional economic activities**						
Yes	228	14.8	**16.0%**	10.8%	**5.940**	**0.015**
No	1316	85.2	84.0%	**89.2%**		

* Percentage in column. Values in bold correspond to the highest values when statistically significant differences exist.

## Data Availability

The data presented in this study are available on request from the corresponding author.

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
