# Peer review of "Rural Ties and Consumption of Rural Provenance Food Products—Evidence from the Customers of Urban Specialty Stores in Portugal"

_foods, 2022, doi:10.3390/foods11040547_

Round 1

Reviewer 1 Report

The paper is very interesting but the manuscript needs to be improved.

Please, consider adding "in Portugal" in the title.

The authors should cite other studies in the first sentence of the Introduction part (Line 36-37).

Please, remove "(" in Line 51.

The authors should write in the Introduction or part 2.2 about the fact that the COVID-19 pandemic makes us reconsider the relevance of short food supply chains and local products (this creates a rationale for the research).

Could the authors avoid the wording "presented above" (Line 236 and others) and "as mentioned" (Line 260)?

The author writes about Figure 2 but there is no Figure 2 in the main text!

I have some questions regarding methodology. How were the survey items developed, i.e. by the authors themselves or by adapting items from other studies? What are the studies? Were the self-developed scales validated?

It would be preferable to include questionnaire in the Appendix.

I suggest to write more about the limitations of the study.

There is no information on ethical approval in the article. Why?

Decision: major revision.

Author Response

We would like to thank the Reviewer for the valuable comments and suggestions. Please find our detailed answers in the file attached and in the resubmitted manuscript. 

Best Regards

Reviewer 2 Report

This is an interesting study exploring the factors affecting consumption of rural provenance food products in Portugal. The study is comparing two different groups frequent and sporadic buyers and tries to identify the factors affecting their consumption of the products under investigation.

Title: One of the main issues I have is whether the sonclusions drawn from this study refer to consumption or buying behaviour of these products. Since all the questions addressed to the participants referred to the reasons they buy the food products or questions on which products they buy. The participants could be customers but not necessarily consumers of the products. Therefore the difference between the two needs to be addressed in the introduction of this particular study (the review of the literature refers to customers and consumers). It also needs to be clear if the sample is referring to customers or consumers. Based on this observation the title will need to be readdressed and not include the word consumption.

Abstract: The abstract summarises the study but in many cases the sentences are too long and it is diffucult to follow the author’s point of view. Specifically:

Line 18-20 the word “which” should be replaced by “of which”, the expression “embodied by” could be “embodied in”

Line 21: “representations” a word used throughout the study I do not feel it represents the meaning of what the authors are measuring. It could be replaced by images as it is used in the tables, or characterisation. If it is decided to continue to be used then it is best to use the singular form.  

Line 26: the authors are using organoleptic and sensorial as two different attributes. These words are considered synonyms therefore the one can be used instead of the two throughout the document.

Lines 28-31 a very long sentence where the point of the authors is very difficult to follow. Moreover, the word contexts could be context

Introduction:

The introduction justifies the purpose of this study but my only concern here would be to discuss the difference between the customers and consumers.

Literature review:

The literature review is well structured with some minor expression issues:

Line 81: replace the word “dialogically” this refers to a dialog and I cannot see the connection here

Line 87: “specialisations” do the authors refer to speciality?

Line 107: “sensorial and organoleptic” use one of the two

Materials and methods

Although the description of the methodology is clear there are a number of issues that need to be addressed.

How were the data on “representations” collected? Were these open-ended questions? Or did the participants had to check all that applied from a list of attributes? If these questions were open-ended how were the data analysed?

Line 240: a pre-test could be replaced with the word pilot test.

Line 241: “input” instead of “inputs”

In terms of the quantitative data which software was used for the analysis?

Line 257: describe which t-test was used?

Results and discussion

Table 1:  replace the word “foreigner” with “not- Portuguese” because it reads less discriminatory

Table 2: it is not clear where the ** only the values corresponding to yes refers to at the note of the table

Could there be discussion around the differences between Portuguese and not- Portuguese consumers because this would add some interesting information? Maybe not for all the factors investigated but for the representations of rural areas, and rural food products.

Conclusions

The conclusions could be discussed in relation to other European countries not only from the Mediterranean region. It would also be important to discuss how the findings of this research could guide not only further research but also product development, policy development and education of consumers. The authors mention sustainability in their introduction therefore this should also be addressed here.

References and citations

Throughout the citations need to be improved so that the name of the authors should be included with the number. For example, in line 339 the authors mention “in line with [41] and [47]” but it should be best presented as “in line with DuPuis et al. [41] and Amilien et al., [47]” This should be corrected throughout the document. Another example is in line 332 “as evidence by [40]”

Author Response

We would like to thank the Reviewer for the valuable comments and suggestions. We have addressed those comments and suggestions in the file attached and in the resubmitted manuscript. 

Best Regards

Round 2

Reviewer 1 Report

The authors have thoroughly revised their article.

Author Response

Dear Reviewer

Thank you very much for your kind comments

Best Regards